# Serum Level of Vitamin D Is Associated with Severity of Coronary Atherosclerosis in Postmenopausal Women

**DOI:** 10.3390/biology10111139

**Published:** 2021-11-05

**Authors:** Ewelina Anna Dziedzic, Wiktor Smyk, Izabela Sowińska, Marek Dąbrowski, Piotr Jankowski

**Affiliations:** 1Medical Faculty, Lazarski University in Warsaw, 02-662 Warsaw, Poland; 2Department of Internal Medicine and Geriatric Cardiology, Centre of Postgraduate Medical Education, 01-813 Warsaw, Poland; piotrjankowski@interia.pl; 3Liver and Internal Medicine Unit, Department of General, Transplant and Liver Surgery, Medical University of Warsaw, 02-091 Warsaw, Poland; wiktorsmyk123@gmail.com; 4Medical Faculty, Medical University of Warsaw, 02-091 Warsaw, Poland; sowinska.izabela@gmail.com; 5Department of Cardiology, Bielanski Hospital, 01-809 Warsaw, Poland; mardab@vp.pl; 6Institute of Cardiology, Jagiellonian University Medical College, 31-008 Krakow, Poland

**Keywords:** 25(OH)D, atherosclerosis, coronary artery disease, Coronary Artery Surgery Study, estrogen deficiency, myocardial infarction, modifiable CAD risk factors

## Abstract

**Simple Summary:**

Postmenopausal women experience rapid progression of coronary artery disease. Recent studies have shown that vitamin D deficiency might be considered a modifiable risk factor for cardiovascular diseases. The main object of this study was to analyze the potential cardioprotective effect of 25(OH)D and its influence on coronary atherosclerosis assessed by Coronary Artery Surgery Study Score (CASSS). We prospectively enrolled over 300 postmenopausal women undergoing coronary angiography. Our study showed that women with more severe coronary atherosclerosis have significantly lower serum levels of 25(OH)D. We also presented that vitamin D serves as an independent determinant of the CASS Score. Our study provides further evidence that low vitamin D level appears to be a potential risk factor for coronary artery disease.

**Abstract:**

**Background:** Postmenopausal women experience rapid progression of coronary artery disease. Vitamin D deficiency appears to be a modifiable risk factor for cardiovascular diseases. This study aimed to analyze the impact of 25-hydroxyvitamin D3 (25(OH)D) level on the severity of coronary atherosclerosis and its potential cardioprotective effect in postmenopausal women. **Material and Methods:** The study prospectively recruited 351 women in postmenopausal age undergoing coronary angiography. The severity of coronary atherosclerosis was assessed using the Coronary Artery Surgery Study Score (CASSS). A level of 25(OH)D was measured with electrochemiluminescence. **Results:** Women with more severe coronary atherosclerosis have significantly lower 25(OH)D serum level (*p* = 0.0001). Vitamin D (β = −0.02; *p* = 0.016), hypertension (β = 0.44; *p* = 0.025), age (β = 0.02; *p* = 0.003), and history of MI (β = 0.63; *p* < 0.0001), were shown as CASSS determinants. Age, hyperlipidemia, and history of MI were found to determine the level of vitamin D (all *p* < 0.05). Women with a three-vessel disease hospitalized due to ACS, with a history of MI, hyperlipidemia and hypertension presented the lowest vitamin D level. **Conclusions:** Our study showed that lower serum 25(OH)D in postmenopausal women is associated with more significant stenosis in the coronary arteries. Therefore, we suggest considering low vitamin D level as a potential risk factor for coronary artery disease.

## 1. Introduction

To date, cardiovascular diseases, including coronary artery disease (CAD), are the most common cause of death in women in Poland and all over the world [1,2]. CAD is present relatively occasionally in young women, who are at a lower risk of cardiovascular events due to the cardioprotective effect of estrogens [3,4]. The risk of CAD increases dramatically with age, after menopause reaching the same incidence as in men [5]. Due to the long-term vascular protection linked with estrogens, women have their first myocardial infarction about 9 years later than men [6]. Additionally, the risk of CVD mortality in women over 65 years of age is significantly higher than in men of the same age [7]. According to Hulley et al. [8], approximately half of women over 50, thus statistically experiencing the onset of menopause (mean menopausal age 48.8 years [9]) can be diagnosed with CAD. Few researchers have addressed the issue of CAD and other CVD as the reason for increased mortality in women with earlier menopause [10,11,12,13].

Due to various factors determining the dissimilar disease course, the distinction of male and female cohorts with coronary artery disease is required. Recent observational studies confirmed that women and men experience chest pain similarly, but it differs in duration, intensity, and radiation. [14,15]. The causes of MI in women include ruptured plaque, microcirculation dysfunction or vasospasm, and spontaneous dissection of the coronary artery. However, women are more likely to experience myocardial infarction in the absence of obstructive coronary arteries (MINOCA) [16]. Therefore, the diagnosis and treatment of ACS in women more appears to be more challenging than in men [17]. Studies have shown that women are less likely to receive optimal drug therapy and are less frequently considered eligible for coronary angiography and coronary revascularization [18,19]. Worse outcomes of percutaneous coronary interventions (PCI) lead to more frequent re-hospitalization rates and often result in repetition of the procedure [20]. In addition, women have higher in-hospital, early and late mortality [21].

It should be noted that most studies assessing the effectiveness of primary CAD prevention may not be representative due to the small percentage of women participating in clinical trials (15–40%) and lack of gender-specific analysis [22,23,24]. Most of the data on IHD is still largely derived from men. Thus, the actual incidence of IHD in women may be underestimated [25,26].

To date, numerous studies investigated the cardioprotective effect of vitamin D and the association between its serum concentration and various cardiovascular conditions i.e., cardiac function after MI, subclinical myocardial injury, hypertension, and CAD progression [27]. In addition to the above-mentioned effect on arterial hypertension, a relationship between calcitriol and other well-established cardiovascular risk factors such as atherogenic lipid profile [28], diabetes [29], obesity [30] has been demonstrated. An inverse relationship between the concentration of 25 (OH) D and various fractions of cholesterol, including low-density cholesterol (LDL-C) has been demonstrated [28].

Vitamin D improves the functions of vascular endothelial cells by protecting against end products of glycosylation [31], increasing the activity of endothelial nitric oxide synthase [32] or reducing the production of reactive oxygen species [32]. In vitro studies suggested that vitamin D may downregulate the NF-kB pathway and actively suppress the transcription of proinflammatory cytokines (IL-6, IL-8, TNF-alfa) [33]. Thus, it could directly attenuate the progression of atherogenesis in coronary arteries [19].

In addition to its anti-inflammatory effect, vitamin D inhibits the processes responsible for the destabilization of the atherosclerotic plaque, such as the transformation of macrophages into foam cells [34] or the formation of new vessels by inhibiting vascular endothelial growth factor (VEGF) [35]. Calcitriol exerts an anticoagulant effect by inhibiting platelet adhesion and increasing the production of thrombomodulin [36]. Taking into account its cardioprotective potential, insufficient serum level of vitamin D might be considered as a modifiable risk factor of cardiovascular diseases [37]. The ad-equate intracellular concentration of 1,25(OH)2-vitamin appeared to reduce the inflammatory response and decrease CAD progression. However, the results of most randomized controlled trials were inconclusive [27].

Investigation of the potentially modifiable risk factors of CAD is highly required considering the increasing prevalence of CVD. Diverse symptomatology and disease course in men and women point out the need for re-examining CAD in the female patient cohort. Therefore, this study aimed to analyze the impact of serum 25-hydroxy vitamin D level on the severity of coronary atherosclerosis and episodes of myocardial infarction in postmenopausal women.

## 2. Materials and Methods

### 2.1. Study Cohort and Clinical Data

This cross-sectional study comprises data of Polish patients referred to the cardiology department for the diagnostic catheter angiography and the evaluation of coronary artery disease in 2016. The study protocol (KB/124/2014) was approved by the local ethics committee, according to the ethical guidelines of the 1975 Declaration of Helsinki (latest revision, 2013), and written, informed consent was obtained from all participants.

The inclusion criteria were gender and age: women over 50 years old. Exclusion criteria were: chronic kidney disease (stage III-V) or accompanying disorders of calcium and phosphate regulation; active malignancy or potential paraneoplastic syndrome and imbalance of calcium and phosphate homeostasis; elevated inflammatory markers (total white blood cell count >10,000 cells/μL or C-reactive protein concentration >5 mg/L) or fever; medication or supplementation containing vitamin D or calcium. In each individual, serum concentration of phosphate, calcium, and parathyroid hormone was measured. Participants with lower and upper laboratory reference limits of serum calcium and were excluded from the study. All patients were treated with comparable doses of atorvastatin or rosuvastatin.

### 2.2. Measurements

A simple questionnaire was used to collect patients’ data and asses smoking habits. All individuals underwent a detailed clinical and biochemical examination including the measurements of body mass index (BMI), serum 25(OH)D level, fasting plasma glucose, total cholesterol (TC), triglycerides (TG), low-density lipoprotein (LDL), high-density lipoprotein (HDL), systolic and diastolic blood pressure. Coronary angiography was performed in each patient to assess the degree of coronary atherosclerosis.

### 2.3. Examinations

Standing height and weight were measured during the physical examination and were used to calculate BMI (kg/m^2^). Blood samples were collected into clot activator tubes from the antecubital vein of fasted subjects. All biochemical blood tests were determined by standard clinical-chemical assays in the laboratory in our centre. The serum concentration of 25(OH)D was determined with the chemiluminescent immunoassay (DiaSorin, Stillwater, MN, USA). The vitamin D status was classified according to the Endocrine Society Clinical Practice Guidelines for Vitamin D Deficiency [38]: 25(OH)D level <10 ng/mL was considered as a severe deficiency, ≥10 to <20 ng/mL as a moderate deficiency, ≥20 to <30 ng/mL as a mild deficiency, and ≥30 ng/mL as optimal. Patients were examined throughout the whole year, and factors that might have influenced 25(OH)D levels were noted. Examination data corresponds to the season of blood draw, which was reported by the National Health and Nutrition Examination Survey (NHANES) as winter months (November to April) and summer months (May to October) [39]. Fasting plasma glucose concentration ≥126 mg/dL or a random plasma glucose measurement was >200 mg/dL were diagnosed as diabetes. Total cholesterol, high-density lipoprotein and triglycerides were measured by an enzymatic method. Low-density lipoprotein concentration was calculated using the Friedewald formula [40]. Hyperlipidemia was defined as TC >200 mg/dL and/or TG > 150 mg/dL. Dyslipidemia was defined as LDL-C ≥ 70 mg/dL, HDL-C < 50 mg/dL. Hypertension was defined as ≥140 mmHg systolic and/or ≥ 90 mmHg diastolic. The results of patients with chronic kidney disease stage >G3 (GFR < 45 mL/1.73 m^2^) were excluded from the statistical analysis due to disturbances of calcium and phosphate metabolism accompanying kidney disease. The coronary angiography was performed in all patients conducted to the study using standard diagnostic catheters through radial or femoral artery access. Coronary angiography is a procedure that uses dye contrast and x-rays to detect blockages in the coronary arteries usually caused by atherosclerotic plaques. The study was performed using standard radial or femoral catheters by a team of three experienced doctors. The degree of coronary narrowing was assessed visually, while the Fractional Flow Reserve (FFR) was used in patients with intermediate-degree coronary artery stenosis. Most of the patients (99.6%) were treated with statins (atorvastatin or rosuvastatin) and only 0.4% has not received this kind of medication. The influence of antihypertensive and antidiabetic drugs was not taken into account, which is a limitation of the present study. Coronary Artery Surgery Study Score (CASSS) was used to assess the degree of coronary atherosclerosis [41]. Stenosis >70% in any of the large epicardial coronary arteries (anterior descending branch, circumflex branch, and right coronary artery) was scored at 1 point. Stenosis ≥50% of the left main coronary artery was scored at 2 points and further considered a two-vessel disease. The score was calculated as the sum of all points. The score may illustrate one, two, or three-vessel CAD [41]. The diagnosis of the acute coronary syndrome (ACS) was based on the current European Society of Cardiology guidelines [42]. The basis of the diagnosis of acute myocardial infarction was an increase in myocardial necrosis markers (especially troponin) and at least one of the following criteria: symptoms of myocardial ischemia, ECG changes indicating recent ischemia, formation of the pathological Q waves on the ECG, evidence in imaging studies of a new loss of viable myocardium or a new segmental disturbance in the movement of the heart wall, a thrombus in a coronary artery in the angiography.

### 2.4. Statistical Analyses

The Shapiro-Wilk test was used to evaluate the data distribution. To compare the results of continuous variables between the two groups, the Mann-Whitney-Test or *t*-test was used. Kruskal-Wallis test or one-way ANOVA were used to compare three or more independent groups for parametric and non-parametric distribution, respectively. Pearson’s chi-squared test or Fisher’s exact test were used to determine differences between prevalence in selected groups. The Poisson regression analysis was used to assess the relationship between the CASS score and selected variables. Multivariable linear regression analysis was performed to assess the relationship between vitamin D level and selected variables. The variance inflation factor was calculated to measure the impact of collinearity among variables in the model. A two-sided *p*-value < 0.05 was considered statistically significant. The statistical analysis was performed with SPSS 26.0 (SPSS, Munich, Germany). Figures were created using GraphPad Prism 8.0 (GraphPad Software, San Diego, CA, USA).

## 3. Results

Detailed baseline characteristics of the study cohort are presented in Table 1. In summary, we enrolled 351 female patients, their median age was 71 years (range 50–92), BMI 28.1 kg/m^2^ (range 16.9–47.8), 66 (18.8%) declared active tobacco smoking. Overall, 112 (31.9%) had diabetes mellitus, 205 (58.4%) had hyperlipidemia, 296 (90.5%) had dyslipidemia, 303 (86.3%) were diagnosed with hypertension. Each enlisted participant underwent successful coronary angiography. The CASSS 0,1,2,3 was confirmed in 132 (37.6%), 89 (25.4%), 73 (20.8%), 57 (16.2%) individuals, respectively. A total of 131 (37.3%) women were admitted to the hospital because of acute coronary syndrome, whereas 220 (62.7%) were hospitalized due to chronic coronary syndromes (CCS).

Severe deficiency (25(OH)D level < 10 ng/mL), moderate deficiency (≥10 to <20 ng/mL), mild deficiency (≥20 to <30 ng/mL) and optimal level (≥30 ng/mL) were observed in 93 (26.5%), 158 (45.0%), 71 (20.2%), 29 (8.3%) patients, respectively.

As presented in Table 1, an analysis of the variance showed statistically significant differences in age (*p* = 0.002), HDL (*p* < 0.0001), serum levels of vitamin D (*p* = 0.0001) between various CASSS values. Figure 1 illustrates the substantial decrease of 25(OH)D level with the increase of CASS score. Patients with a CASSS 0 have a significantly higher concentration of serum 25(OH)D as compared to those with a CASSS 1, 2, or 3 (*p* < 0.01 for all comparisons). Figure 2 presents a significant decline of serum HDL level with the increment of atherosclerosis progress in coronary arteries, reflected by CASSS. Women without significant changes in coronary vessels (CASSS 0) have a significantly higher concentration of HDL compared with those with CASSS 1, 2, or 3 (*p* < 0.01 for all comparisons). Moreover, a statistically significant association was found between the prevalence of hypertension (*p* = 0.02), MI in history and ACS (both *p* < 0.0001) and the severity of atherosclerosis of coronary arteries.

Multiple regression analysis of Poisson-distributed data was used to evaluate factors that may determine CASS score in a group of women over 50 years of age. The following determinants were included in the final analysis: age, BMI, serum 25(OH)D level, TC, LDL, HDL, TG, diabetes, hypertension, hyperlipidemia, history of myocardial infarction, and season during the examination. As presented in Table 2, statistically significant determinants of CASSS were: age (*p* = 0.003), 25(OH)D (*p* = 0.016), hypertension (*p* = 0.025), and history of MI (*p* < 0.0001).

The regression model was designed to verify the determinants of vitamin D level in postmenopausal women. The analysis showed (Table 3) that age (*p* = 0.03), hyperlipidemia (*p* = 0.01), and history of MI (*p* = 0.02) significantly influenced the concentration of 25(OH)D.

Identification of the most vulnerable subgroup of patients showed that women with CASSS 3, hospitalized due to ACS, with a history of MI, hyperlipidemia and hypertension had the lowest vitamin D level in the entire cohort.

To evaluate both protective and risk factors for ACS, we included all previously investigated factors as covariates into the logistic regression model. Results are presented in Table 4. Univariate regression analysis demonstrated significant (all *p* < 0.05) associations between ACS and age, 25(OH)D, HDL and smoking status. Inclusion of significant factors from the univariate into multivariate analysis revealed that age, HDL and smoking status are independent determinants of ACS.

## 4. Discussion

In the present study, we demonstrated a link between low serum vitamin D levels and the severity of coronary atherosclerosis in postmenopausal women. Patients with the more pronounced atherosclerosis in coronary vessels (i.e., higher CASSS) had a significantly lower concentration of 25(OH)D. The most vulnerable subgroup of women (i.e., CASSS 3, hospitalized for ACS, MI in history, hyperlipidemia, and arterial hypertension) presented the lowest level of vitamin D. The findings of our current research are in line with previous studies that involved the association of serum calcitriol concentration and CVD [43,44].

Vitamin D deficiency accompanied by the growing incidence of cardiovascular events generates interest of the researchers [45,46,47]. To date, a low serum level of 25(OH)D is associated with both higher incidence and increased CVD mortality [48,49]. Several studies have suggested the possible protective role of vitamin D on the cardiovascular system [50,51]. However, most of these studies were based on both gender groups with male predominance. The main objective of our current research was to assess the influence of serum vitamin D level on the severity of CAD reflected by CASS score and myocardial infarction episodes in postmenopausal women. To our knowledge, it is one of the first studies to assess the relationship between these two variables in this group of patients. Taking into account major differences in gender-pattern of the disease course and rapid disease progression after menopause, we aimed to provide additional data on potentially modifiable CAD risk factors in female patients.

The influence of vitamin D deficiency on episodes of MI is supported by several cohort studies [48,52]. They showed that the threshold of 25(OH)D <15 ng/mL is associated with a 1.5- to 2-fold higher risk of MI. As suggested by Ng et al., low vitamin D level and a history of MI may significantly increase the risk of further major adverse cardiovascular events (MACE), including re-occurrence of MI [53]. Noteworthy, serum 25(OH)D level above 7.3 ng/mL reduced the risk of non-fatal MACE by 40% in patients with ACS.

Previous studies assessing the link between serum 25(OH)D level and the severity of CAD were performed in cohorts of different nationalities and the participation of female individuals reached up to 40% [54,55,56,57]. Therefore, they cannot be directly compared with the results obtained within this study. In 2015, Verdoia et al. investigated women of varying ages and underlined the relationship between the lower level of vitamin D and both incidence of CAD and severity of coronary atherosclerosis [43]. On the other hand, Morgan et al. analyzed women over 40 years of age and presented a significant inverse correlation between serum 25(OH)D level and the degree of coronary vessels stenosis assessed in coronary angiography [44]. Within this cohort, coronary artery stenosis >50% occurred over 2 times more frequently in individuals with severe deficiency of vitamin D (<10 ng/mL) as compared with those with concentration >20 ng/mL [44]. Interestingly, patients receiving daily supplementation of 0.5 μg calcitriol for six months had significantly reduced inflammation of the coronary arteries and thus, significantly declined SYNTAX score [58]. However, large cohort studies did not confirm the favourable effect of vitamin D supplementation on cardiovascular outcomes [59].

We are aware that several limitations of our study need to be addressed. The study cohort comprised exclusively women living in the central part of Poland. The classification of the severity of atherosclerosis was based on coronary angiography images and the scale that does not recognize the calcifications in coronary vessels, which stabilize atherosclerotic plaques. All study participants were treated with a statin, without taking into account the dose and duration of treatment. Menopausal status was not confirmed by the diagnostics of cessation of ovarian hormonal activity. The inclusion criterium in the study was age over 50 years old, which exceeds the median age of menopause onset [9,60]. The study was observational and cross-sectional; consequently, it cannot prove causation but demonstrate a statistical association.

The latest scientific discoveries emphasize the multidirectional action of vitamin D and its significant participation in the pathomechanisms of many disorders, including cardiovascular diseases. The observational studies cited above have confirmed the association of low serum VD concentration with higher mortality due to cardiovascular events. The results of our work suggest an association of vitamin D deficiency in both with the severity of coronary atherosclerosis as well as its complications in the form of ACS in the group of postmenopausal women. In the analyzed group of patients, due to the lack of estrogen, the lower synthesis of the active form of VD, it seems that supplementation of this substance in higher doses may be indicated mainly to overcome the increased activity of the parathyroid glands. The negative metabolic consequences of calcitriol deficiency presented in the manuscript indicate the need to prevent or ultimately treat the deficiency of this compound. Therefore, we suggest considering vitamin D deficiency as an easily modifiable risk factor for CAD in the group of postmenopausal patients. In our opinion, well-designed campaigns including education about proper exposure to solar radiation, or pharmacological supplementation could be notably beneficial. As there is still no confirmation of the relationship between vitamin D deficiency and the risk of cardiovascular events, randomized clinical trials are necessary.

## 5. Conclusions

In conclusion, we presented that lower serum 25(OH)D in postmenopausal women is associated with significant stenosis in the coronary arteries. Therefore, we suggest considering vitamin D as a potential risk factor for coronary artery disease. Since the availability of data on postmenopausal women is limited, further studies are required to assess the final impact of calcitriol on atherosclerotic plaque and the potential beneficial effect of its supplementation in such patients.

## Figures and Tables

**Figure 1 biology-10-01139-f001:**
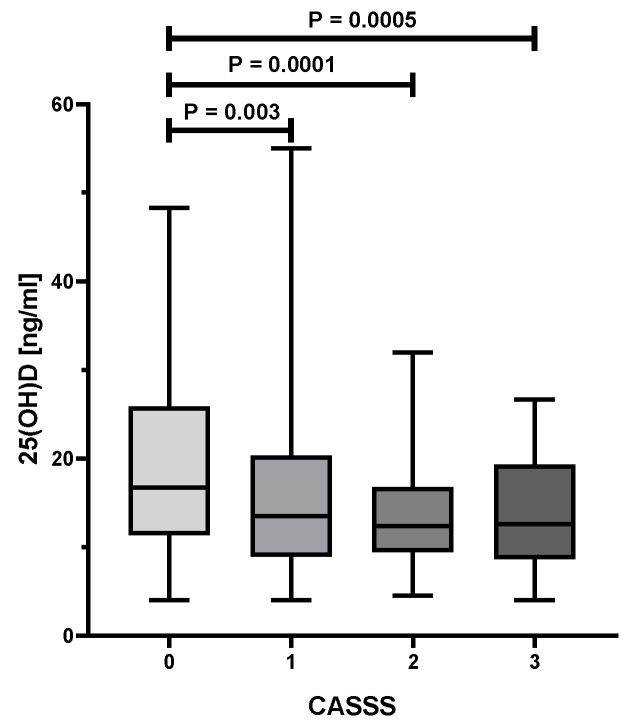
Differences between vitamin D level and Coronary Artery Surgery Study (CASS) score in postmenopausal women.

**Figure 2 biology-10-01139-f002:**
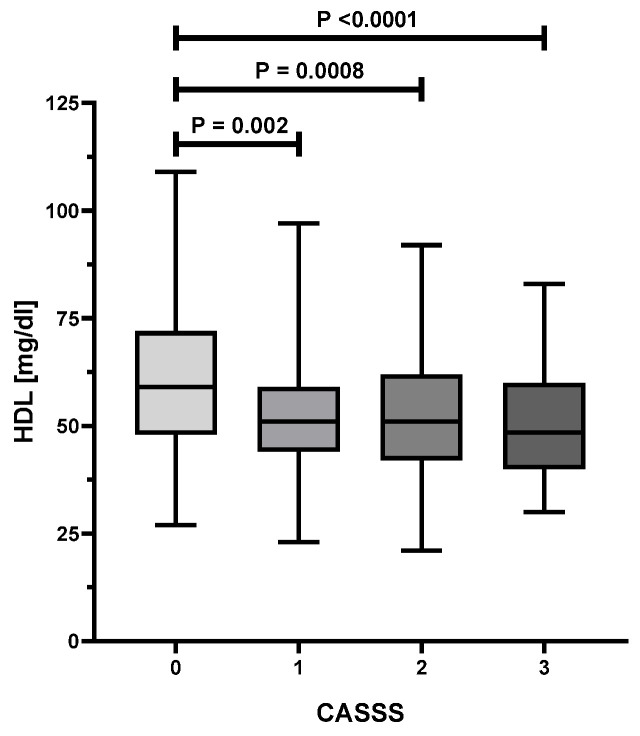
Association between HDL and CASS score.

**Table 1 biology-10-01139-t001:** Baseline characteristics of the study cohort.

		CASS Score				
	Entire Cohort	0	1	2	3	*p*
Total participants, n	351	132 (37.6%)	89 (25.4%)	73 (20.8%)	57 (16.2%)	
Age (years)	71 (50–92)	69 (52–91)	71 (51–92)	73 (52–87)	75 (52–91)	0.002
Smoking, n	66 (18.8%)	20 (15.1%)	24 (27.0%)	15 (20.5%)	7 (12.3%)	0.18
BMI [kg/m^2^]	28.1 (16.9–47.8)	28.2 (19.2–44.8)	26.8 (16.9–47.8)	28.5 (17.4–38.4)	28.4 (18.3–43.3)	0.46
Hyperlipidemia, n	205 (58.4%)	69 (52.3%)	52 (58.4%)	45 (61.6%)	39 (68.4%)	0.19
Dyslipidemia, n	296 (90.5%)	112 (84.9%)	71 (79.8%)	61 (83.6%)	52 (91.2%)	0.42
Total cholesterol [mg/dL]	181.8 (70.9–367.0)	182.2 (85.4–366.3)	179.8 (110.0–351.6)	188.7 (70.9–338.3)	178.0 (84.8–310.8)	0.66
Triglycerides [mg/dL]	15.6 (31.3–386.3)	115.5 (31.3–281.0)	110.8 (48.7–357.6)	18.7 (54.0–386.3)	112.2 (46.9–306.4)	0.50
HDL [mg/dL]	52.3 (21.3–54.8)	58.8 (26.8–109.0)	50.6 (23.5–97.4)	51.0 (21.3–92.1)	48.6 (29.7–82.6)	<0.0001
LDL [mg/dL]	101.6 (20.5–289.2)	94.5 (20.5–253.0)	105.2 (39.7–289.2)	105.7 (24.4–235.0)	101.9 (27.3–228.3)	0.83
Serum 25(OH)D [ng/mL]	14.0 (4.0–55.0)	16.7 (4.0–48.3)	13.5 (4.0–55.0)	12.4 (4.5–32.0)	12.6 (4.0–26.7)	0.0001
Diabetes mellitus, n	112 (31.9%)	33 (25.0%)	29 (32.6%)	25 (34.2%)	25 (43.9%)	0.07
Hypertension, n	303 (86.3%)	108 (81.8%)	74 (83.1%)	65 (89.0%)	56 (98.2%)	0.02
MI in history, n	111 (31.6%)	9 (6.8%)	33 (37.0%)	36 (49.3%)	33 (57.9%)	<0.0001
Acute coronary syndrome, n	131 (37.3%)	16 (12.1%)	51 (57.3%)	33 (45.2%)	31 (54.4%)	<0.0001

Abbreviations: 25(OH)D, 25-hydroxyvitamin D; BMI, Body mass index; CASS, Coronary Artery Surgery Study; HDL, high-density lipoprotein; MI, myocardial infarction; LDL, low-density lipoprotein.

**Table 2 biology-10-01139-t002:** Determinants of CASS score.

	β	95% CI	OR (95% CI)	*p*
Age	0.02	0.01–0.03	1.02 (1.01–1.03)	0.003
BMI	−0.01	−0.04–0.01	0.99 (0.97–1.01)	0.25
25(OH)D	−0.02	−0.03–−0.01	0.98 (0.97–1.00)	0.016
Total cholesterol	0.03	−0.07–0.13	1.03 (0.93–1.14)	0.58
LDL	−0.03	−0.13–0.07	0.97 (0.88–1.07)	0.58
HDL	−0.04	−0.14–0.06	0.96 (0.87–1.06)	0.40
Triglycerides	−0.01	−0.03–0.02	1.00 (0.98–1.02)	0.60
Diabetes mellitus	0.13	−0.11–0.37	1.14 (0.90–1.44)	0.28
Hypertension	0.44	0.06–0.82	1.55 (1.06–2.28)	0.025
Hyperlipidemia	0.20	−0.12–0.51	1.22 (0.89–1.67)	0.21
Dyslipidemia	0.30	−0.09–0.69	1.35 (0.92–2.00)	0.13
MI in history	0.63	0.41–0.85	1.81 (1.51–2.35)	<0.0001
Smoking status	0.05	−0.24–0.34	1.05 (0.78–1.41)	0.76
Season during the examination	0.01	−0.25–0.27	1.01 (0.78–1.31)	0.94

Abbreviations: 25(OH)D, 25-hydroxyvitamin D; BMI, Body mass index; CASS, Coronary Artery Surgery Study; Cl—confidence interval; HDL, high-density lipoprotein; MI, myocardial infarction; OR, odds ratio; LDL, low-density lipoprotein.

**Table 3 biology-10-01139-t003:** Determinants of vitamin D level in women over 50 years of age.

	Univariate		Multivariate					
	β	*p*	β	VIF	*p*	β	VIF	*p*
Age	−0.07	0.21	−0.13	1.17	0.03	−0.13	1.16	0.03
BMI	−0.03	0.63	0.002	1.18	0.97	0.00	1.17	0.99
Total cholesterol	−0.13	0.02	−0.43	398.17	0.69	NA	NA	NA
LDL	−0.15	<0.01	0.30	312.99	0.76	NA	NA	NA
HDL	0.13	0.02	0.21	40.63	0.54	0.07	1.32	0.24
Triglycerides	−0.15	<0.01	0.01	21.14	0.99	−0.09	1.53	0.21
Diabetes mellitus	−0.03	0.64	0.05	1.21	0.40	0.05	1.20	0.46
Hypertension	−0.01	0.07	−0.08	1.12	0.19	−0.08	1.10	0.18
Hyperlipidemia	−0.19	<0.01	−0.10	2.15	0.20	−0.16	1.26	0.01
Dyslipidemia	−0.08	0.16	0.03	1.36	0.77	0.004	1.30	0.95
MI in history	−0.14	0.01	−0.13	1.11	0.02	−0.13	1.13	0.02
Smoking status	−0.04	0.49	−0.04	1.18	0.54	−0.04	1.17	0.54
Season during the examination	−0.06	0.25	−0.09	1.04	0.12	−0.09	1.03	0.13

Abbreviations; see Table 2; VIF, Variance inflation factor.

**Table 4 biology-10-01139-t004:** Risk and protective factors for ACS.

	Univariate Analysis		Multivariate Analysis	
Factor	OR (95% CI)	*p*	OR (95% CI)	*p*
AGE	1.03 (1.01–1.06)	0.02	1.04 (1.02–1.07)	0.003
BMI	1.00 (0.96–1.05)	0.96	-	-
25(OH)D	0.97 (0.95–1.00)	0.03	0.99 (0.96–1.02)	0.38
TOTAL CHOLESTEROL	1.00 (0.99–1.00)	0.36	-	-
LDL	1.00 (0.99–1.01)	0.54	-	-
HDL	0.97 (0.95–0.98)	<0.0001	0.97 (0.95–0.99)	<0.0001
TRIGLYCERIDES	1.00 (0.99–1.00)	0.43	-	-
DIABETES MELLITUS	1.37 (0.87–2.17)	0.18	-	-
HYPERTENSION	1.19 (0.62–2.26)	0.61	-	-
HYPERLIPIDEMIA	1.26 (0.79–2.01)	0.33	-	-
DYSLIPIDEMIA	1.30 (0.59–2.85)	0.52	-	
SMOKING STATUS	1.77 (1.04–3.02)	0.04	2.31 (1.26–4.24)	0.007

Abbreviations: see Table 2.

## Data Availability

Data can be provided by the authors upon reasonable request.

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
