# Peer review of "Serum Level of Vitamin D Is Associated with Severity of Coronary Atherosclerosis in Postmenopausal Women"

_biology, 2021, doi:10.3390/biology10111139_

Round 1
Reviewer 1 Report
To clarify the association between serum 25(OH)D and coronary arterial disease among postmenopausal women, this author conducted a cross-sectional study of 352 postmenopausal women who undergoing coronary angiography. And this author found that lower serum 25(OH)D in postmenopausal women is associated with more significant stenosis in the coronary arteries. Aa a reviewer, I thought this manuscript contains newly informative knowledge. However, I also found there are many maters that should be solved.
(Abstract)
1) In abstract, spell out “25(OH)D” is necessary.
2) In abstract, the magnitude of the present main correlations should be shown. The readers could not image the importance of present findings without those data.
3) In abstract, even this author described about the association between vitamin D and same variables, this author did not describe about vitamin D in conclusion.
(Introduction)
4) The main points of the description are not have been clarified. Even present study is aimed to clarify the association between 25(OH)D and coronary artery score among postmenopausal women, the description of 25(OH)D is limited. Therefore, readers could not understand the biological characteristics of 25(OH)D enough.
5) There are many contents that are not directly related to present study. Present description is focused on the importance of myocardial infraction prevention too much. However, present study is not interventional study that is intended to prevent myocardial infarction.
(Material and Methods)
6) What methods to measure present variables should be clarified. According to present description, LDL-cholesterol was measured. Is this expression correct? This study measured total-cholesterol, HDL-cholesterol, and triglycerides. Many studies used LDL-cholesterol values by calculated form total-cholesterol, HDL-cholesterol, and triglycerides.
7) Detail description of coronary angiography also should be shown. The number of examiner and the experience of examiner also could be informative.
8) This author used 10ng/mL 20ng/mL and 30ng/mL as cut-off values for 25(OH)D. However, to void the impact of arbitrary, quartile values should be used.
9) Even present study has the data of HDL and LDL, why not use dyslipidemia as a confounding factor on present model.
10) How is the influence of medication for hypertension, dyslipidemia, diabetes on present study?
11) Renal function also could be influence on present results. Since renal condition might affect both on concentration of 25(OH)D [Ref1] and atherosclerosis, renal function also could act as an important confounding factor on present study.
[Ref1] Pekkarinen T, et al. The same annual dose of 29000 IU of vitamin D (cholecalciferol) on either daily of four monthly basis for elderly women: 1-year comparative study of the effects on serum 25(OH)D concentrations and renal function. Clin Endocrinol (Oxf). 2010;72(4):455-61.
12) Details of the definition of ACS also should be described in text.
(Results)
13) As shown in table 2 and table 3, this author performed multivariable liner regression analysis. However, before performing multivariable liner regression model, linear association among major findings should be clarified. Then plots that clarify the main correlations should be shown as figures.
14) In addition to standardized parameter estimates for present results as be shown in table 2 and 4, simple correlation coefficient for each variable should be shown.
15) What variables were used in multi-adjusted model should be clarified in text and footnote of table 4.
(Discussion)
16) First paragraph of discussion section is not appropriate description for discussion section.
17) The major findings of present study should be clarified the first paragraph of present study.
18) What is the newly informative knowledge in present study should be shown in discussion section with same references.
19) Clinical implication for present study also should be described.
20) Since this is the cross-sectional study, causal relationship could not be established.
Author Response
Dear Reviewer
Thank you very much for reviewing the manuscript entitled "Serum Level Of Vitamin D Is Associated With Severity Of Coronary Atherosclerosis In Postmenopausal Women". We hope that our manuscript, after introducing changes in accordance with the comments received, now presents much more scientific value. The received guidelines allowed us to correct the presented work. The introduced changes are presented below point by point:
1) In abstract, spell out “25(OH)D” is necessary.
Answer: According to the reviewer's suggestion, we added to the abstract the 25(OH)D abbreviation meaning.
2) In abstract, the magnitude of the present main correlations should be shown. The readers could not image the importance of present findings without those data.
Answer: According to the reviewer’s suggestion, we have reorganized the abstract, the magnitude of the correlations is shown.
3) In abstract, even this author described about the association between vitamin D and same variables, this author did not describe about vitamin D in conclusion.
Answer: The abstract was corrected according to the reviewer’s suggestion.
4) The main points of the description are not have been clarified. Even present study is aimed to clarify the association between 25(OH)D and coronary artery score among postmenopausal women, the description of 25(OH)D is limited. Therefore, readers could not understand the biological characteristics of 25(OH)D enough.
Answer: The introduction section was improved according to the reviewer’s suggestion.
Page 3-4, lines: 97-120
5) There are many contents that are not directly related to present study. Present description is focused on the importance of myocardial infraction prevention too much. However, present study is not interventional study that is intended to prevent myocardial infarction.
Answer: According to the reviewer’s suggestion we crossed out most content related to the prevention of MI.
6) What methods to measure present variables should be clarified. According to present description, LDL-cholesterol was measured. Is this expression correct? This study measured total-cholesterol, HDL-cholesterol, and triglycerides. Many studies used LDL-cholesterol values by calculated form total-cholesterol, HDL-cholesterol, and triglycerides.
Answer: Total cholesterol, high-density lipoprotein and triglycerides were measured by an enzymatic method. Low-density lipoprotein concentration was calculated using the Friedewald formula.
On the revised manuscript we added this information: Page 5, lines 177-179.
7) Detail description of coronary angiography also should be shown. The number of examiner and the experience of examiner also could be informative.
Answer: Thank you for pointing this out. We added this topic to the material and methods section. This is included in the lines 185-194 of the revised manuscript.
8) This author used 10ng/mL 20ng/mL and 30ng/mL as cut-off values for 25(OH)D. However, to void the impact of arbitrary, quartile values should be used.
Answer: In the presented study, we use the classification of vitamin D levels recommended in polish endocrine guidelines [RusiÅ„ska A., PÅ‚udowski P., Walczak M. i wsp.: Vitamin D supplementation guidelines for general population and groups at risk of vitamin D deficiency in Poland – recommendations of the Polish Society of Pediatric Endocrinology and Diabetes and the Expert Panel with Participation of National Specialist Consultants and Representatives of Scientific Societies – 2018 Update. Front. Endocrinol. (Lausanne), 2018; 9: doi: 10.3389/fendo.2018.00 246]. We agree with the reviewer and agree that the cut-points we used are arbitrary. However, we decided to use the cut-points related to the commonly used classification of vitamin D deficiency. According to the Reviewer’s suggestion we are going to use the quartile values in the subsequent studies. We hope the reviewer will accept our explanation.
9) Even present study has the data of HDL and LDL, why not use dyslipidemia as a confounding factor on present model.
Answer: As required by the reviewer, we added dyslipidemia in all performed analyses and its diagnostic criteria in the material and methods section.
10) How is the influence of medication for hypertension, dyslipidemia, diabetes on present study?
Answer: The effect of drugs on high blood pressure or diabetes was not considered in the study, and this was noted in the limitations of the study. The patients enrolled in the study were treated with a statin (atorvastatin or rosuvastatin). The appropriate explanation was added to the material and methods and discussion section.
11) Renal function also could be influence on present results. Since renal condition might affect both on concentration of 25(OH)D and atherosclerosis, renal function also could act as an important confounding factor on present study.
Answer We agree that this is a potential limitation however, patients with chronic kidney disease stage> G3 (GFR <45 ml / 1.73 m2) were excluded from the statistical analysis due to disturbances of calcium and phosphate metabolism accompanying kidney disease. We hope the reviewer will accept our explanation.
12) Details of the definition of ACS also should be described in text.
Answer: According to the reviewer’s suggestion, we present ACS diagnostic criteria in the material and methods section.
13) As shown in table 2 and table 3, this author performed multivariable liner regression analysis. However, before performing multivariable liner regression model, linear association among major findings should be clarified. Then plots that clarify the main correlations should be shown as figures.
Answer: Since CASS Score is count data and has Poisson distribution, the Poisson regression analysis was conducted – results are presented in Table 2. To assess the relation between selected determinants and CASS Score, analysis of variance was performed – results of these analyses are presented in Table 1. In Table 3 results of multivariable linear regression analysis are shown. Linear association among selected determinants and vitamin D level is now presented using variance inflation factor.
14) In addition to standardized parameter estimates for present results as be shown in table 2 and 4, simple correlation coefficient for each variable should be shown.
Answer: Variance inflation factor (VIF) was calculated to measure the impact of collinearity among variables in the presented model.
15) What variables were used in multi-adjusted model should be clarified in text and footnote of table 4.
Answer: On p. 8 of the previously submitted manuscript, we included the criteria for inclusion into the multivariate regression model.
16) First paragraph of discussion section is not appropriate description for discussion section.
Answer: The discussion section was corrected according to the reviewer’s suggestion.
17) The major findings of present study should be clarified the first paragraph of present study.
Answer: The first paragraph of the discussion in the revised manuscript has been updated as suggested by the reviewer.
18) What is the newly informative knowledge in present study should be shown in discussion section with same references.
Answer: The discussion section was improved according to the Reviewer’s suggestion. Page 9.
19) Clinical implication for present study also should be described.
Answer:
The latest scientific discoveries emphasize the multidirectional action of vitamin D and its significant participation in the pathomechanisms of many disorders, including cardiovascular diseases. The observational studies cited above have confirmed the association of low serum VD concentration with higher mortality due to cardiovascular events. The results of our work suggest an association of vitamin D deficiency in both with the severity of coronary atherosclerosis as well as its complications in the form of the acute coronary syndrome in the group of postmenopausal women. In the analyzed group of patients, due to the lack of estrogen, the lower synthesis of the active form of VD, it seems that supplementation of this substance in higher doses may be indicated mainly in order to overcome the increased activity of the parathyroid glands. The negative metabolic consequences of calcitriol deficiency presented in the manuscript indicate the need to prevent or ultimately treat the deficiency of this substance. Therefore, we suggest considering vitamin D deficiency as a new, easily modifiable risk factor for coronary artery disease in the group of postmenopausal patients. In our opinion, well-designed campaigns including education in the field of exposure to solar radiation or pharmacological supplementation will bring the greatest benefit. As there is still no clear confirmation of the causal relationship between vitamin D deficiency and the risk of cardiovascular events, randomized clinical trials are necessary.
20) Since this is the cross-sectional study, causal relationship could not be established.
Answer: We added the following text at the end of the discussion containing the limitations of the presented study: The study was observational and cross-sectional; consequently, it cannot prove causation but demonstrate a statistical association.

Reviewer 2 Report
This paper described circulating levels of VitD in relation to the cardiovascular risk, in particular atherosclerosis, in postmenopause women. This study is single time-point and descriptive study in nature. As the authors stated, their major finding is that in the postmenopausal women, lower VitD level is associated with increased risk of coronary arteries.
Whilst the association of low VitD levels and higher cardiovascular risk has been reported by numerous studies, I regard two findings here that are potentially important. First, the authors indicated in the methods part that they performed coronary artery angiogram to evaluate the degree of CAD in this cohort in relation to the VitD level. However, the paper was unclear on the findings from CA angiogram, otherwise demonstrating the degree of coronary artery stenosis as well as number of arteries in relation to VitD levels would be new and important. Second, a few factors, such as age, hypertension, remote history of myocardial infarction and dyslipidemia are identified to influence VitD levels in this cohort. This observation is useful in the interpretation of VitD data.
Comments
- There is insufficient presentation of the data derived from coronary angiogram. Were all subjects undergone coronary angiogram? What was the lumen stenosis or the number of arteries involved revealed by CA angiogram? This reviewer had difficulty in locating these basic CAD data.
- This reviewer has concern on statistics used. The data were analysed mainly on single variant basis. In fact, multivariant logistic regression analysis is the suitable statistical method to ascertain that VitD level is indeed an independent risk factor. This is especially important as you showed that the level of VitD is influenced by several factors as indicated above. Without such more vigorous statistical analysis, it would be impossible to establish a causal role of lower VitD level and CAD.
Author Response
Dear Reviewer
Thank you for giving us the opportunity to submit a revised draft of the manuscript "Serum Level Of Vitamin D Is Associated With Severity Of Coronary Atherosclerosis In Postmenopausal Women". A comprehensive summary of our manuscript and insightful comments let us introduce valuable corrections. Below please find a point-by-point response to editor comments.
- There is insufficient presentation of the data derived from coronary angiogram. Were all subjects undergone coronary angiogram? What was the lumen stenosis or the number of arteries involved revealed by CA angiogram? This reviewer had difficulty in locating these basic CAD data.
The coronary angiography was performed in all patients conducted to the study using standard diagnostic catheters through radial or femoral artery access. Coronary angiography is a procedure that uses dye contrast and x-rays to detect blockages in the coronary arteries usually caused by atherosclerotic plaques. The study was performed using standard radial or femoral catheters by a team of three experienced doctors. The degree of coronary narrowing was assessed visually, while the Fractional Flow Reserve (FFR) was used in patients with intermediate-degree coronary artery stenosis. Coronary Artery Surgery Study Score (CASSS) was used to assess the degree of coronary atherosclerosis [33]. Stenosis >70% in any of the large epicardial coronary arteries (anterior descending branch, circumflex branch, and right coronary artery) was scored at 1 point. Stenosis ≥50% of the left main coronary artery was scored at 2 points and further considered a two-vessel disease. The score was calculated as the sum of all points. The score may illustrate one, two, or three-vessel CAD [33].
- This reviewer has concern on statistics used. The data were analysed mainly on single variant basis. In fact, multivariant logistic regression analysis is the suitable statistical method to ascertain that VitD level is indeed an independent risk factor. This is especially important as you showed that the level of VitD is influenced by several factors as indicated above. Without such more vigorous statistical analysis, it would be impossible to establish a causal role of lower VitD level and CAD.
We improved our statistical approach. CASS Score is count data and has Poisson distribution, consequently the Poisson regression analysis was conducted – results are presented in Table 2. To assess relation between selected determinants and CASS Score, analysis of variance was performed – results of these analyses are presented in Table 1. In Table 3 results of multivariable linear regression analysis are shown. Linear association among selected determinants and vitamin D level is now presented using variance inflation factor. Moreover, we performed uni- and multivariate analyses. Importantly, our study is observational and cross-sectional; consequently, it cannot prove causation but demonstrate an association.

Round 2
Reviewer 2 Report
The authors have addressed my concerns. I have no further comment to this work.